# Heightened immigration enforcement impacts US citizens' birth outcomes: Evidence from early ICE interventions in North Carolina

**Romina Tome**[¤a], **Marcos A. Rangel** ⓘ**, Christina M. Gibson-Davis** ⓘ*, **Laura Bellows**[¤b]

Sanford School of Public Policy, Duke University, Durham, North Carolina, United States of America

¤a Current address: American Institutes for Research, Washington, DC, United States of America
¤b Current address: Curry School of Education & Human Development at the University of Virginia, Charlottesville, Virginia, United States of America
* cgibson@duke.edu

**Data Availability Statement:** The data underlying the results presented in the study cannot be shared publicly, as the owner of the data, the North Carolina Department of Vital Statistics, prohibits

## Abstract

We examine how increased Immigration and Customs Enforcement (ICE) activities impacted newborn health and prenatal care utilization in North Carolina around the time Section 287(g) of the Immigration and Nationality Act was first being implemented within the state. Focusing on administrative data between 2004 and 2006, we conduct difference-in-differences and triple-difference case-control regression analysis. Pregnancies were classified by levels of potential exposure to immigration enforcement depending on parental nativity and educational attainment. Contrast groups were foreign-born parents residing in nonadopting counties and all US-born non-Hispanic parents. The introduction of the program was estimated to decrease birth weight by 58.54 grams (95% confidence interval [CI], −83.52 to −33.54) with effects likely following from reduced intrauterine growth. These results are shown to coexist with a worsening in the timing of initiation and frequency of prenatal care received. Since birth outcomes influence health, education, and earnings trajectories, our findings suggest that the uptick in ICE activities can have large socioeconomic costs over US-born citizens.

## Introduction

Over the last decade, immigration enforcement has increased dramatically in the US interior, with a large number of individuals experiencing detention and removal [1,2]. These actions by the Immigration and Customs Enforcement (ICE) agency have impacted the psychological, social, and economic stresses faced by immigrant families, with potential deleterious effects over physical health [3,4]. Pregnant women are particularly vulnerable to these stressors, insofar as mothers-to-be, and their fetuses, are vulnerable to the hardening of the environment in which they live. Given that conditions during pregnancy partly determine adult health and well-being [5–7], ICE activity experienced during gestation may have long-term consequences for the health and well-being of generations of US citizens.

One of the main instruments of this increase in immigration enforcement were federal-local partnerships, known as 287(g) programs. Under 287(g) programs, local law officers are

the sharing of birth record data that contains geographic identifiers. However, other researchers wishing to obtain the data may submit requests to the North Carolina Department of Vital Records (https://vitalrecords.nc.gov/contacts.htm). The authors had no special access privileges to the data and other researchers will be able to access the data in the same manner as the authors.

**Funding:** M.R. & C.G.D. Robert Wood Johnson Foundation www.rwjf.org The funders had no role in study design, data collection and analysis, decision to publish, or preparation of the manuscript.

**Competing interests:** The authors have declared that no competing interests exist.

deputized to act as ICE agents. In practice, these partnerships give local law officers the authority and discretion to question individuals about immigration status, and if necessary, begin deportation proceedings [8,9]. The 287(g) programs were reinvigorated under the Trump administration, and, in response to an executive order signed by the president upon taking office in January 2017; particularly towards the end of 2019, programs across the country have more than doubled in number [10]. Currently 139 jurisdictions, located mostly in southeast states or states along the southern US border, have active 287(g) programs [11].

Insofar as 287(g) programs both reflect and intensify state- and local-level animus towards non-citizens, it is reasonable to consider that they potentially harm the health of all immigrants (not just those at risk of deportation). Immigrant access to health care and other social goods heavily depends on the integration or criminalization of immigrants in the states and communities in which they live [8,9,12–17]. We, therefore, examine the hypothesis that ICE activity, as manifested through 287(g) programs, likely heightens the risk of adverse birth outcomes for immigrant mothers, either by increasing levels of maternal stress, leading to maladaptive pregnancy behaviors, by curtailing access to medical care, or by adversely affecting economic conditions.

Increasing levels of maternal stress are harmful as stress is manifested physiologically through spikes in glucocorticoids (cortisol) levels, which have programming effects on the behavioral and physiological development of children affected while in utero [18–26]. Maladaptive coping mechanisms, such as increasing cigarette consumption [27,28], have serious implications for the health and well-being of the fetus [29,30]. Access to medical care may be reduced as, under increased enforcement, immigrant mothers are less likely to enroll in public insurance, receive public benefits, or obtain adequate prenatal care [31–33]. ICE activity also leads to "chilling effects", where feelings of fear, anxiety, and mistrust prevent immigrants from seeking out services that they otherwise would be entitled to [15,34]. Finally, increases in immigration enforcement may adversely impact labor market participation and household economic conditions. Families experiencing a detention or removal typically lose family income [35–37], with immigration enforcement being thought to reduce employment rates, particularly among less-educated men [38].

Though previous work has not quantitatively analyzed the repercussions of specific ICE policies like 287(g) programs, studies have demonstrated that immigration enforcement activities can have deleterious effects on Hispanic immigrant mothers and their children. An immigration raid in Iowa led to increased risk of low birth weight [39], whereas legislation in Arizona mandating immigrant documentation caused average birth weights to decline (the law was signed, but never enacted) [40]. In addition to focusing on birth weight to the exclusion of other maternal health behaviors, these studies were limited in that they did not analyze a specific, existing ICE policies that favor universal enforcement and discretionary police action.

We complement these studies by estimating the impact of 287(g) programs on birth outcomes and prenatal care utilization. 287(g) programs are currently in use by immigration authorities, but our study focuses on the retrospective analysis of the 2006 introduction of 287(g) programs in Mecklenburg county, North Carolina. In the mid-2000s, North Carolina was estimated to have 325,000 unauthorized immigrants, the eighth largest unauthorized population by state [41]. These immigrants (primarily from Mexico) clustered in Mecklenburg County (home to the state's most populous city, Charlotte) [42]. Mecklenburg was at the forefront of immigration enforcement: it was not only the first county in the south to sign a 287(g) agreement, but its agreement also served as the model for subsequent 287(g) agreements [43]. The ICE activities in Mecklenburg County in 2006 thus parallels the most recent policy approach to forceful anti-immigrant activities, with potential lessons as to the repercussions of recent ICE operations.

Departing from previous contributions [31,37,40], we focus on families where both the mother and the father are foreign-born and neither parent has a high school diploma. Less-educated immigrants, and men in particular, are the groups most likely to experience removal or fear of removal [44]. The exposure of families (and not just women) to ICE's activity more accurately reflects the experience of immigrant families and the babies they have conceived.

## Methods

### Population and data

Data are long-form birth certificate data from the North Carolina Detailed Birth Records (NCDBR) database. The NCDBR encompasses all North Carolina births and contains information on parental demographics, infant health, and geographic identifiers. The latter includes parents' county of residence and country of birth. The NCDBR does not contain information on parental immigration documentation status (we are not aware of any large set of administrative data on births that does).

North Carolina is an appropriate setting for this study for a number of reasons. First, the state has become a new destination for immigrants, with a 13-fold increase in the number of unauthorized immigrants between 1990 and 2017. In 2017, the state had an unauthorized population of 325,000, the eighth largest in the country [41]. Relatedly, the number of births to foreign-born mothers in North Carolina has soared, and approximately one in five births in the state is to a foreign-born mother. Additionally, when Mecklenburg County adopted the policy in 2006, it was the first instance in the country of the so-called universal enforcement approach of the 287(g) program, in which alleged noncitizens could be asked about their authorization status regardless of their criminal records [43]. Data from Syracuse University's Transactional Records Access Clearinghouse (TRAC) project indicates that 39% of the detainers (i.e., documents that allow local law enforcement agencies to hold immigrants until they can be placed in removal proceedings) issued in the state between 2003 and 2009 were originated in Mecklenburg County. More than 84% of the detainers in that county were for minor violations of the law (e.g., traffic related), and 96% of those detained were Hispanic.

The sample was constructed in two steps. In the first step, to guarantee representation of the foreign-born maternal population as a percentage of the population and within the calendar year, we limited the sample to births in counties that met two restrictions: (1) at least 5% of births between 2004 and 2005 were to foreign-born mothers, and (2) at least one birth in every month observed was to a foreign-born mother. These two restrictions provide the minimum conditions to avoid empty cells within the analysis while maximizing the number of counties included. Of North Carolina's 100 counties, 47 counties met these restrictions. We present the list of counties in S1 Table. Second, among those 47 counties, we limited the sample to births that occurred in the nine months immediately after 287(g) was implemented (March–November 2006) and births that occurred during the same time frame one year earlier (March–November 2005). This time frame provides a consistent (and seasonality adjusted) window of months both before and after 287(g) was implemented. Note that restricting the sample to births in the nine months after the adoption of the program focuses our analysis on fetuses likely conceived prior to the implementation of the 287(g) program (confirmed using self-reported information on conception date). By using this time-window restriction, our findings are shielded from potential behavioral responses in the form of fertility decisions.

The main portion of our sample consists of observations on singleton live births to foreign-born parents subdivided by level of education: parents with less than high school education ($n$ = 12,588) and parents with high school education or more ($n$ = 8,397), as well as by county of maternal residence. We conduct our analyses by contrasting such sample with observations

consisting of 96,018 live births to non-Hispanic US-born parents during the same period and within the same locations. Hispanic US-born parents were likely affected by the anti-immigrant sentiment and risk of profiling faced by the broader Hispanic community [39,45]. When we included Hispanic US-born parents in the comparison group, results were not significantly different (albeit smaller) from the impacts we report below. This suggests statistically insignificant spillover effects of immigration policies over the small population of Hispanic-American parents in North Carolina.

## Key measures

**Birth outcomes and maternal prenatal care use.** We examined two sets of outcomes: child health at birth and access to care. The first group of outcomes were based on birth weight and small-for-gestational age status. Birth weight was measured both continuously in grams and dichotomously using an indicator for low birth weight ($<$ 2,500 grams). Being small for gestational age was defined as membership in the smallest decile of birth weight, by week of gestation, calculated using nationally defined curves for fetal growth [46].

Maternal adequacy of prenatal care utilization is measured through the Kotelchuck index [47], a dichotomous indicator of whether a woman initiated care prior to the fourth month of pregnancy and received at least 50% of recommended visits. Smoking and alcohol use during pregnancy were also considered as outcomes but were rejected as they were extremely rare in our sample: less than 0.5% of mothers in our main sample report smoking or consuming alcohol while pregnant.

**Other covariates.** As covariates, we included child gender (1 = female), whether the child was firstborn, maternal age, maternal race/ethnicity, and maternal marital status (1 = married at birth, 0 = otherwise). We controlled for maternal age using indicator variables to avoid making any assumptions about functional form. Maternal race and ethnicity included non-Hispanic white, non-Hispanic Black, Hispanic, and other race and ethnicity. The sample was not limited to Hispanics, as the 287(g) program targeted all unauthorized people, regardless of ethnicity. We note, however, that time-invariant racial/ethnic differences should not bias our estimates, since our research design relies solely on similarities of temporal variation within each subgroup.

## Analytical strategy

We used difference-in-differences models to estimate the effect of the 287(g) program. Difference-in-differences models estimate changes in outcomes for a treatment group before and after a policy is introduced with changes in outcomes for a control group that is not subject to the policy. In our case, we compared neonatal health (as an example outcome) in Mecklenburg County before and after the 287(g) program was implemented with neonatal health from children from the same demographic group residing around the same time in counties that did not adopt the program. The method relies on the assumption that, in the absence of the 287(g) program, changes in neonatal health among those most likely affected by the 287(g) program within Mecklenburg County would be equivalent to changes in neonatal health among similar children in the rest of the state. We estimated difference-in-differences models on three subpopulations: foreign-born parents without a high school diploma, foreign-born parents with a high school diploma or more, and non-Hispanic US-born parents. All analyses were conducted using STATA (version 15; StataCorp).

Formally, we considered alternative versions of the following model:

$$Y_{ict} = \beta_0 + \beta_1 Meck_{ic} + \beta_2 After_{it} + \beta_3 Meck_{ic} After_{it} + u_{ict} \qquad (1)$$

where $Y_{ict}$ indicates an outcome for newborn $i$ whose parents reside in county $c$ at the time of the child's birth, $t$. $Meck_{ic}$ is an indicator that takes the value 1 if the child's parents reside in Mecklenburg County when the child is born and 0 if they live in another county. $After_{it}$ takes value 1 if child $i$ is born after Mecklenburg County adopted the 287(g) program—that is, after February 27, 2006—and 0 if the child was born before the implementation. Our coefficient of interest is $\beta_3$. If, for example, $Y$ is birth weight, $\beta_3$ indicates that, relative to children born in other counties, the growth in average weight in Mecklenburg County were $\beta_3$ grams higher around the passage of the 287(g) program. This relative increase would then be attributed to the new policy. In subsequent models, we added to the above by including county fixed effects, thereby controlling for all county-specific characteristics that do not change over time, and year-month fixed effects, thereby accounting for period-specific state-level policies or shocks. We used robust standard errors, clustered at the county level, to construct 95% CIs.

To provide a validity check, we also estimated the difference-in-differences model among less-educated foreign-born parents but including only births in the years 2004 and 2005 (predating the actual policy implementation) while assuming that the 287(g) program had been implemented one year before it actually was. The difference-in-differences model relies on the parallel-trends assumption between control and treatment groups. Therefore, detecting that the passage of time impacts these groups differentially before the 287(g) program is implemented would cast doubt on our inferences regarding the actual policy enactment. Incidentally, the exact same estimation can be used to indicate if individuals could have anticipated the implementation of the 287(g) program and adapted their behaviors before the administration of the treatment (by possibly being exposed to media coverage of the incoming policy change). In other words, significant results in the model using the 2004–2005 data would cast doubt on a policy-induced impact on outcomes, insofar as differences in outcomes would have followed temporal patterns that pre-date policy being studied here or were significantly deflated by anticipatory behaviors.

Next, we estimated the impact of potential coincidental events that could have affected Mecklenburg County but not the other counties in the state. To do so, we conducted triple-difference estimation. In triple-difference models, estimates from separate difference-in-differences models are compared to check if they significantly differ from each other (in essence, the difference in the difference-in-differences estimator for two groups). We conducted two alternative triple-difference estimations. The first compared the difference-in-differences estimates for less-educated foreign-born parents to the difference-in-differences estimates for better-educated foreign-born parents. The second compared the difference-in-differences estimates for our sample of interest, less-educated foreign-born parents, to the difference-in-differences estimate for non-Hispanic US-born parents. Significant estimates in the triple-difference model would indicate (for example) that the change in neonatal health for children of less-educated foreign-born parents in Mecklenburg County differed from the neonatal health change for children of better-educated foreign-born parents who also resided in Mecklenburg County and gave birth around the same time. The model is as follows:

$$Y_{ict} = \delta_0 + \delta_1 NEDI_{ic} + \delta_2 Meck_{ic}After_{it} + \delta_3 Meck_{ic}NEDI_{ic} + \delta_4 After_{it}NEDI_{ic}$$
$$+ \delta_5 Meck_{ic}After_{it}NEDI_{ic} + \phi_c + \phi_t + X_{ict} + \varepsilon_{ict}$$

(2)

where $NEDI_{ic}$ takes the value 1 if child $i$'s mother is less educated and both parents are foreign-born. Our coefficient of interest is $\delta_5$, which indicates the differential effect of immigration enforcement on these groups of parents. This is equivalent to taking the difference between the difference-in-differences estimator for two groups of children. Our model includes county

fixed effects ($\phi_c$), year-by-month fixed effects ($\phi_t$), and additional covariates ($X_{ict}$) capturing child (gender, birth order) and maternal (age, marital status, and race) characteristics.

We also conducted analyses in which we employed different control groups. These groups included counties that adopted 287(g) programs after 2006, counties that applied for 287(g) programs but were denied, counties that either adopted 287(g) programs or applied for them after 2006, or counties that neither adopted nor tried to adopt 287(g) programs.

While some researchers utilize the staggered adoption of a policy across locations to jointly estimate a difference-in-differences parameter, we did not do so here. Our reasoning is that 287(g) programs are inherently local, and likely not comparable across locations–after all, the important element is the discretion given to local authorities. Counties signing agreements with ICE after Mecklenburg County did so by adapting from previous experiences within the state and under a different party-orientation of federal offices. Moreover, each county may have not experienced the same path of treatment effects considering that they differ in their covariates, which affect the response to treatment. As a growing set of technical working papers has been pointing out, violating the assumption of homogeneity of treatment would lead to non-robust difference-in-differences estimates [e.g., 48–50].

## Results

Descriptive statistics highlight differences between mothers in Mecklenburg County and mothers in other parts of the state, for one year before the implementation of the program and the nine months afterward (Table 1). Notably, in both periods, regardless of county, Hispanics were overrepresented within the group of less-educated parents among foreign-born parents. Differences by county of residence among foreign-born parents were minimal except that in both periods, less-educated foreign-born parents in Mecklenburg County, relative to similar parents in other counties, were less likely to be married and more likely to be on their first birth. These differences are stable over time.

A parallel set of descriptive statistics for outcomes (Table 2) indicates that prior to the implementation of the 287(g) program (top panel), less-educated foreign-born parents in Mecklenburg County, relative to similar parents in other North Carolina counties, had comparable birth outcomes. Similar geographic comparisons with better-educated foreign-born parents, and with non-Hispanic US-born parents, likewise indicate no statistically significant differences in birth outcomes in the pre-287(g) period. Utilization of prenatal care for less-educated foreign-born, however, was lower in Mecklenburg County relative to similar parents elsewhere even before the policy change we evaluate. Since our empirical strategy is based on time-changes, this difference in levels is unlikely to invalidate our results and conclusions, as further discussed below.

After the policy was implemented (bottom panel), birth outcomes for less-educated foreign-born parents in Mecklenburg County worsened: infants weighed less and were more likely to be small for gestational age. Parents were also even more likely to report inadequate prenatal care utilization. Differences within Mecklenburg County for the other subgroups of parents in birth outcomes were not found, though better-educated foreign-born parents in Mecklenburg County also reported reduced rates of adequate access to prenatal care.

The negative impact of the policy on less-educated foreign-born mothers is formalized through difference-in-differences models (Table 3, first to third columns). The reduction in birth weight likely induced by the 287(g) program ranged from a raw estimate of 59.34 grams (95% CI, −83.26 to −35.42) among less-educated foreign-born parents to 58.02 grams (95% CI, −82.45 to −33.59) after the full set of control variables are included. Significantly worsened outcomes can also be seen in terms of small-for-gestational-age births (2.25 percentage points,

**Table 1. Descriptive statistics on maternal demographics for births to foreign- and US-born parents residing in North Carolina (March to November, 2005 and 2006).**

| | Births to less-educated foreign-born parents (N = 12,588)[a] | | | Births to more-educated foreign-born parents (N = 8,397) | | | Births to non-Hispanic US-born parents (N = 96,018) | | |
|---|---|---|---|---|---|---|---|---|---|
| | Meck. | Other NC[b] | P Value[c] | Meck. | Other NC[b] | P Value[c] | Meck. | Other NC[b] | P Value[c] |
| **Pre-287(g)[d]** | | | | | | | | | |
| Maternal age, y, mean | 25.57 | 25.98 | 0.04 | 29.26 | 29.22 | 0.84 | 29.49 | 27.62 | 0.00 |
| Mother Hispanic, % | 95.28 | 96.81 | 0.05 | 38.53 | 38.10 | 0.81 | NA | NA | NA |
| Mother white non-Hispanic, % | 1.30 | 1.20 | 0.82 | 16.84 | 17.01 | 0.90 | 66.92 | 76.18 | <0.001 |
| Mother Black non-Hispanic, % | 0.47 | 0.30 | 0.49 | 14.67 | 14.07 | 0.65 | 32.47 | 21.64 | <0.001 |
| Mother less educated, % | NA | NA | NA | NA | NA | NA | 6.48 | 11.25 | <0.001 |
| Married at birth, % | 42.03 | 49.22 | <0.001 | 79.96 | 81.97 | 0.17 | 74.79 | 72.69 | <0.001 |
| First live birth, % | 30.81 | 26.33 | 0.01 | 38.95 | 40.38 | 0.43 | 44.43 | 42.58 | 0.01 |
| Female child born, % | 50.53 | 48.81 | 0.35 | 49.79 | 48.12 | 0.37 | 49.91 | 48.57 | 0.63 |
| **Post-287(g)[e]** | | | | | | | | | |
| Maternal age, y, mean | 25.93 | 26.16 | 0.24 | 29.33 | 29.47 | 0.42 | 29.38 | 27.48 | 0.00 |
| Mother Hispanic, % | 94.80 | 96.74 | 0.01 | 42.24 | 37.70 | 0.01 | NA | NA | NA |
| Mother white non-Hispanic, % | 1.08 | 1.09 | 0.98 | 12.78 | 15.85 | 0.01 | 64.86 | 75.79 | <0.001 |
| Mother Black non-Hispanic, % | 1.30 | 0.36 | 0.01 | 12.60 | 12.42 | 0.88 | 34.09 | 22.17 | <0.001 |
| Mother less educated, % | NA | NA | NA | NA | NA | NA | 6.85 | 11.26 | <0.001 |
| Married at birth, % | 35.54 | 49.12 | <0.001 | 79.50 | 81.85 | 0.09 | 72.67 | 71.06 | 0.01 |
| First live birth, % | 31.64 | 25.32 | <0.001 | 39.84 | 39.72 | 0.94 | 44.35 | 43.32 | 0.14 |
| Female child born, % | 49.40 | 48.75 | 0.71 | 50.22 | 48.65 | 0.36 | 48.72 | 48.60 | 0.87 |

Abbreviations: Meck., Mecklenburg; NC, North Carolina; NA, not applicable.

[a] Less-educated foreign parents have less than high school (or nonreported) education.

[b] Other North Carolina counties correspond to 46 distinct units.

[c] The P values for the difference in means between treatment and control counties are based on standard errors clustered at the county level.

[d] March to November 2005.

[e] March to November 2006.

95% CI, 0.89 to 3.61). We also detected significant differences in the timely initiation and frequency of prenatal care, which together contributed to an increase of 9.90 percentage points (95% CI, 7.71 to 12.09) in the incidence of births without adequate prenatal care in Mecklenburg County. As expected, when we replicated the analysis without taking fathers into account, we found smaller size effects. For example, for the sample of 17,884 less-educated foreign-born mothers, we observed a reduction of 36.33 grams (95% CI, −56.46 to −16.19; results presented in S2 Table).

Among better-educated foreign-born parents (Table 3, fourth column), the policy did not impact birth outcomes. Interestingly, our estimations indicate that this group also had a significant increase in inadequate medical care during the prenatal phase (although effects are smaller in magnitude than among the less educated). The policy did not impact outcomes among US-born parents (Table 3, fifth column).

Results of the falsification exercise assuming that the policy was enacted in 2005 are presented in the first column in Table 4. No statistical impacts in outcomes were detected; thus, we found no evidence that would suggest differential trends preceding the policy implementation. This finding lends support to the validity of our estimation strategy also with respect to the presence of anticipatory response. In addition, the last four columns in Table 4 show that our findings are robust to the use of different county-control groups.

**Table 2. Descriptive statistics on birth outcomes and pregnancy conditions for births to foreign and US-born parents residing in North Carolina (March to November, 2005 and 2006).**

| | Births to less-educated foreign-born parents (N = 12,588)[a] | | | Births to more-educated foreign-born parents (N = 8,397) | | | Births to non-Hispanic US-born parents (N = 96,018) | | |
|---|---|---|---|---|---|---|---|---|---|
| | Meck. | Other NC[b] | P Value[c] | Meck. | Other NC[b] | P Value[c] | Meck. | Other NC[b] | P Value[c] |
| **Pre-287(g)[d]** | | | | | | | | | |
| Birth weight, g | 3,339.55 | 3,343.18 | 0.85 | 3,307.14 | 3,324.93 | 0.39 | 3,331.06 | 3,326.25 | 0.57 |
| Low birth weight, % | 4.96 | 4.92 | 0.97 | 5.27 | 4.91 | 0.66 | 6.36 | 6.88 | 0.13 |
| Small for gest. age, % | 8.50 | 8.25 | 0.81 | 10.85 | 9.79 | 0.35 | 8.31 | 8.53 | 0.58 |
| Inadeq. prenatal care, % | 27.41 | 24.12 | 0.05 | 10.77 | 10.15 | 0.59 | 4.57 | 5.67 | <0.001 |
| **Post-287(g)[e]** | | | | | | | | | |
| Birth weight, g | 3,298.07 | 3,361.04 | <0.001 | 3,312.97 | 3,323.18 | 0.59 | 3,322.12 | 3,318.55 | 0.66 |
| Low birth weight, % | 5.09 | 4.49 | 0.44 | 5.50 | 5.05 | 0.56 | 6.52 | 6.88 | 0.30 |
| Small for gest. age, % | 10.62 | 7.90 | 0.01 | 11.71 | 10.07 | 0.13 | 8.56 | 8.77 | 0.60 |
| Inadeq. prenatal care, % | 36.80 | 23.78 | <0.001 | 15.19 | 9.84 | <0.001 | 5.92 | 6.04 | 0.72 |

Abbreviations: Meck., Mecklenburg; NC, North Carolina; gest., gestational; Inadeq., Inadequate.

[a] Less-educated foreign-born parents have less than high school (or nonreported) education.

[b] Other North Carolina counties correspond to 46 distinct units.

[c] The P values for the difference in means between treatment and control counties are based on standard errors clustered at the county level.

[d] March to November 2005.

[e] March to November 2006.

Finally, triple-difference estimations (Table 5) indicate that outcomes of children born to less-educated foreign-born parents in Mecklenburg County after the policy change were negatively impacted. Babies born to US-born parents in the same location and around the same time did not have worse outcomes. Taking the difference between the difference-in-differences results for less-educated foreign-born parents and for (all) non-Hispanic US-born parents, the

**Table 3. Change in birth outcomes and health care utilization over time within county of residence, by nativity and education.**

| | Difference-in-Differences Estimates [95% CI][a] | | | | |
|---|---|---|---|---|---|
| | Births to less-educated foreign-born parents (N = 12,588)[b] | | | Births to more-educated foreign-born parents (N = 8,397) | Births to non-Hispanic US-born parents (N = 96,018) |
| | Unadjusted | Adjusted-FE[c] | Adjusted FE-DEM[d] | Adjusted FE-DEM | Adjusted FE-DEM |
| Birth weight, grams | −59.34 | −57.99 | −58.02 | 10.88 | 0.52 |
| | [−83.26, −35.42] | [−81.82, −34.16] | [−82.45, −33.59] | [−15.03, 36.78] | [−8.236, 9.267] |
| Low birth weight | 0.56 | 0.55 | 0.42 | −0.02 | 0.14 |
| | [−0.20, 1.33] | [−0.23, 1.32] | [−0.38, 1.23] | [−0.85, 0.81] | [−0.21, 0.49] |
| Small for gestational age | 2.47 | 2.36 | 2.25 | 0.54 | −0.04 |
| | [1.15, 3.80] | [1.04, 3.67] | [0.89, 3.61] | [−0.67, 1.75] | [−0.42, 0.33] |
| Inadequate prenatal care | 9.73 | 9.83 | 9.90 | 4.58 | 0.90 |
| | [7.40, 12.07] | [7.54, 12.13] | [7.71, 12.09] | [2.16, 7.01] | [0.56, 1.25] |

Abbreviations: FE, fixed effects; DEM, demographics.

[a] The 95% confidence intervals within brackets are based on standard errors clustered at the county level.

[b] Less-educated parents have less than high school (or nonreported) education.

[c] Adjusted FE model includes county fixed effects and month-year fixed effects.

[d] Adjusted FE-DEM model includes county fixed effects, month-year fixed effects, and demographic controls (listed in Table 1).

**Table 4. Change in birth outcomes and health care utilization over time within county of residence, by nativity and maternal education: Robustness checks[a].**

| | DD: births to less-educated foreign-born parents[b] | DDD: less-educated foreign-born vs. US-born parents | | | |
| --- | --- | --- | --- | --- | --- |
| | Counterfactual: policy enactment in 2005 (N = 11,613) | Control counties: future 287(g) adopters (N = 48,488) | Control counties: application for 287(g) denied (N = 28,043) | Control counties: future 287(g) adopters or applicants (N = 63,459) | Control counties: nonadopters and nonapplicants (N = 58,190) |
| Birth weight, g | −13.65 | −74.21 | −81.91 | −74.52 | −39.09 |
| | [−31.85, 4.55] | [−130.48, −17.95] | [−118.00, −45.83] | [−111.11, −37.92] | [−67.49, −10.70] |
| Low birth weight | −0.04 | 1.322 | 0.41 | 0.99 | −0.56 |
| | [−0.75, 0.67] | [−0.20, 2.85] | [−1.35, 2.16] | [−0.02, 1.99] | [−1.76, 0.64] |
| Small for gestational age | 0.14 | 2.75 | 2.06 | 2.44 | 2.07 |
| | [−1.12, 1.40] | [−0.44, 5.93] | [−1.29, 5.41] | [0.29, 4.58] | [0.43, 3.71] |
| Inadequate prenatal care | 0.85 | 9.44 | 7.72 | 9.01 | 8.78 |
| | [−1.72, 3.42] | [5.28, 13.59] | [2.01, 13.43] | [5.95, 12.06] | [5.30, 12.26] |

Abbreviations: DD, difference-in-differences model; DDD, triple-difference model.

[a] Results come from adjusted models that includes county fixed effects, month-year fixed effects, and demographic controls (listed in Table 1). The 95% confidence intervals within brackets are based on standard errors clustered at the county level.

[b] Less-educated foreign-born parents have less than high school (or nonreported) education.

policy reduced birth weight by 58.54 grams (95% CI, −83.52 to −33.54), increased the incidence of small-for-gestational-age births by 2.29 percentage points (95% CI, 0.92 to 3.66) at the same time that inadequate utilization of prenatal care has increased by 8.99 percentage points (95% CI, 6.84 to 11.14). Differences also emerged between the less-educated and better-educated foreign-born parents.

## Discussion

Our study documents negative impacts of an early iteration of 287(g) programs on birth outcomes and prenatal care usage among foreign-born mothers in the Southeastern United States.

**Table 5. Change in birth outcomes and health care utilization over time by nativity and education.**

| | Triple-Difference Estimates [95% CI][a] | |
| --- | --- | --- |
| | Less-educated foreign-born vs. non-Hispanic US-born parents (N = 108,606)[b] | Less- vs. more-educated foreign-born parents (N = 20,985) |
| | Adjusted FE-DEM[c] | Adjusted FE-DEM |
| Birth weight, g | −58.54 | −68.89 |
| | [−83.52, −33.54] | [−104.78, −33.00] |
| Low birth weight | 0.28 | 0.44 |
| | [−0.50, 1.07] | [−0.71, 1.60] |
| Small for gestational age | 2.29 | 1.71 |
| | [0.92, 3.66] | [0.07, 3.35] |
| Inadequate prenatal care | 8.99 | 5.32 |
| | [6.84, 11.14] | [2.71, 7.92] |

Abbreviations: FE, fixed effects; DEM, demographics.

[a] The 95% confidence intervals within brackets are based on standard errors clustered at the county level.

[b] Less-educated foreign-born parents have less than high school (or nonreported) education.

[c] Adjusted FE-DEM model includes county fixed effects, month-year fixed effects, and demographic controls (listed in Table 1).

The children in our study were US-born and therefore citizens by birth. The 287(g) programs are currently in operation, and our study documents that an immigration policy where enforcement discretion is enacted at a local level has adverse effects on infant health.

Our results indicate that the introduction of the 287(g) program reduced birth weight by 58.54 grams and increased the incidence of small-for-gestational-age births by 2.29 percentage points. To contextualize the magnitude of the effect of the 287(g) program on fetal health, we compare our findings against estimates from changes in maternal nutrition, a well-known driver of birth outcomes. Participation in the Supplemental Nutrition Assistance Program (SNAP, formerly known as Food Stamps) has been shown to lead to increases in birth weight of 15 grams to 40 grams among treated women [51], while the Special Supplemental Nutrition Program for Women, Infants, and Children (WIC) has been linked to birth weight improvements of 18 grams to 29 grams [52]. Relative to these impacts, our estimates suggest that the implementation of 287(g) had effects among less-educated immigrants that were larger than the beneficial effect of participating in SNAP or WIC for among overall eligible U.S. population.

Lower birth weight has been related to worse outcomes later in life. Other studies have suggested that a 2% increase in birth weight leads to an increase of 11.5 centimeters in adult height, 0.2 percentage-point in the probability of high school completion, and 0.2 percent in full-time earnings [53]. Lower birth weight has also been linked to worse self-reported health and measures of socio-economic status in adulthood [54]. The incidence of SGA has also been related to to poorer performance at school and lower income. For instance, at age 13, SGA children perform worse than those born appropriate for gestational age (AGA) (with differences of 0.10 and 0.08 in average z-scores for numerical and verbal test respectively) [55]; they are also more likely to be recommended for special education (with differences of 3% at age) [56]. Adults who were SGA report height deficits (-0.55 vs 0.08 of a standardized measure), to be less likely to work in a professional or managerial job (8.7% vs 16.4%) and significantly lower levels of weekly income (11%) than adults who were AGA [56].

The negative impacts estimated here were concentrated among less-educated foreign-born parents, a group that is most likely targeted by immigration enforcement. We argue that effects are concentrated among less-educated parents for two main reasons. First, given assortative mating, women without high school diplomas are likely to have partners with similar levels of education [57]. Insofar as less-educated individuals are more likely than better-educated individuals to be unauthorized [41] and therefore directly subject to 287(g) programs' deportation risk. Less-educated mothers may be more aware of this targeting and experience greater levels of stress when fathers are potential targets. Second, even if they are equally impacted by stress and anti-immigrant sentiment in the location where they reside, better-educated women are more likely to have access to health-protective economic resources.

Finally, consistent with past research [31], we find that 287(g) program's enactment is accompanied by a large reduction on the utilization of adequate prenatal care. This suggests that, under these conditions, increased immigration enforcement may induce among immigrants a reluctance to engage with medical institutions or, equivalently, may experience a decrease in resources that impedes their access to quality prenatal care. In the North Carolina context, there are reports of sheriff's deputies waiting outside migrant health clinics to conduct their immigration-related searches [33]. In fact, although federal policy designates hospitals as sensitive locations, at which immigration enforcement actions are only supposed to be carried out under exigent circumstances, the American Medical Association has called for this designation to be expanded to include spaces within 1,000 feet of any medical treatment or medical facility and to apply under all conditions. Our results suggest the need for closer consideration of remedial initiatives like this.

## Limitations

Limitations to our study should be noted. Our data set, like all administrative data sets of which we are aware, lacks information on documentation status and precluded us from estimating effects on unauthorized immigrants. Based on previous work [58], we assume less-educated women are more likely to be undocumented, but realize education is not a perfect proxy for documentation status.

We also recognize (and addressed in our empirical analysis) two types of unobserved data that could bias our findings downward. First, we do not know if families worked in Mecklenburg County but resided in another county nearby. These families who were included in our control group may have been affected by the implementation of the 287(g) program. Nonetheless, when we conducted estimations using alternative samples of comparison counties (some of which also excluded all counties surrounding Mecklenburg), we found small differences relative to the main specification. Second, we do not observe any media coverage that preceded the passage of 287(g) program, which may have increased fear and stress among Hispanic parents before the policy change, and suggest that we underestimated the true impact of the program. As described above, though, our pre-trends analysis reveals no significant changes in outcomes of interest immediately preceding the policy adoption, casting doubt on this source of bias.

Finally, another limitation of our analysis (and indeed of all studies of immigration enforcement) is that families may have moved geographically in response to policy changes [34,59]. The effect of bias arising from migration on our estimates is unclear, however. If individuals who migrated away from Mecklenburg County were the most economically marginalized, then our analyses would capture an effect that is smaller (in absolute value) than what would have occurred without migration. On the other hand, if those who migrated away were the more economically advantaged (as might be required for changing locations), then our treatment population would change in composition, leading to an overestimation of effects. Though we cannot measure migration, we hypothesize that its effects are small. Fig 1 provides

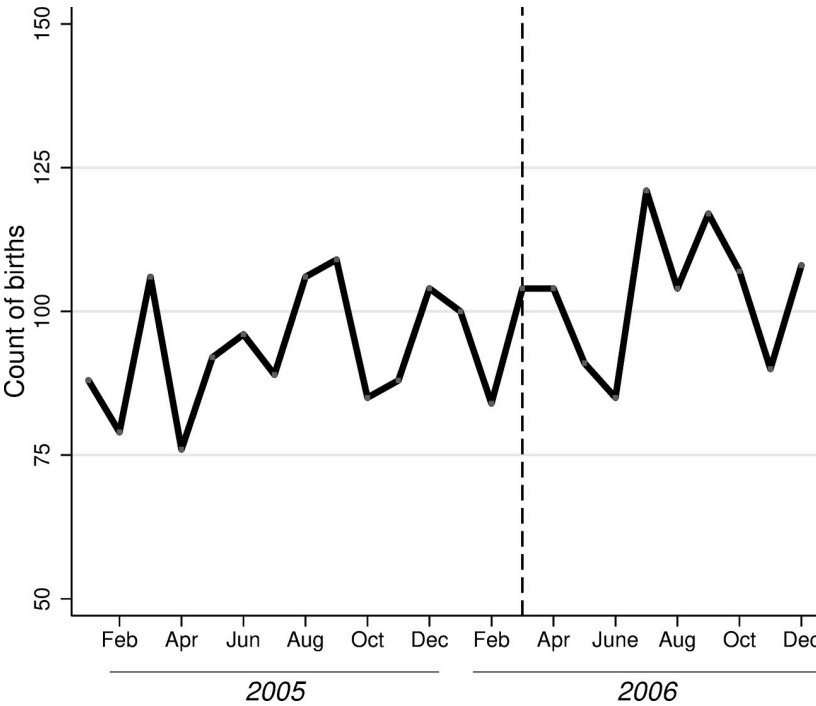

**Fig 1. Monthly count of births for less-educated foreign-born parents in Mecklenburg County (2005–2006).**

indirect support for this argument. If migration were a concern, we would expect to see a decline in the number of births in Mecklenburg County after the introduction of the 287(g) program. However, Fig 1 shows that over time the monthly number of births does not change significantly for less-educated foreign-born parents, and seasonal fluctuations follow those seen in previous year. These findings may be explained by the fact that in the 2000s, Hispanic migrants were clustered in a few large cities in North Carolina [42], likely minimizing the appeal of moving to other locations. We also suspect that pregnant mothers are less likely to change locations than are nonpregnant mothers (since our window of examination surrounds conception). These reasons mitigate our concerns about migration, particularly in relation to other studies based on nonpregnant individuals in locations with more entrenched immigrant populations [60].

## Conclusion

Our results indicate that the introduction of 298(g) programs, which increased local law enforcement's discretion over immigration issues, led to worse health outcomes at birth, with reduced or delayed access to health care likely playing a potentially important mediating role. Our findings, given that birth outcomes heavily influence health, education, and earnings trajectories [5–7], suggest that current socioeconomic costs of the recent uptick in ICE activities under a conservative-leaning federal and local governments' mentality can be long lasting and have deleterious effects on US-born citizens.

## Supporting information

**S1 Table. List of counties included in the analysis.**
(DOCX)

**S2 Table. Change in birth outcomes and health care utilization over time within county of residence, by mother's nativity and education.**
(DOCX)

## Acknowledgments

We thank participants of the Interdisciplinary Association for Population Health Science Conference (2018), and are grateful for feedback received form technical panels of the Robert Wood Johnson's Evidence for Action initiative.

## Author Contributions

**Conceptualization:** Romina Tome, Marcos A. Rangel, Christina M. Gibson-Davis, Laura Bellows.

**Data curation:** Romina Tome, Marcos A. Rangel, Christina M. Gibson-Davis.

**Formal analysis:** Romina Tome, Marcos A. Rangel.

**Funding acquisition:** Marcos A. Rangel, Christina M. Gibson-Davis.

**Investigation:** Romina Tome, Christina M. Gibson-Davis.

**Methodology:** Romina Tome, Marcos A. Rangel.

**Project administration:** Marcos A. Rangel.

**Supervision:** Marcos A. Rangel, Christina M. Gibson-Davis.

**Writing – original draft:** Romina Tome, Marcos A. Rangel, Christina M. Gibson-Davis, Laura Bellows.

**Writing – review & editing:** Romina Tome, Marcos A. Rangel, Christina M. Gibson-Davis.

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
