## [Decision Letter · Decision Letter 0]

27 Oct 2020

PONE-D-20-30120

Heightened immigration enforcement impacts US citizens’ birth outcomes

PLOS ONE

Dear Dr. Gibson-Davis,

Thank you for submitting your manuscript to PLOS ONE. After careful consideration, we feel that it has merit but does not fully meet PLOS ONE’s publication criteria as it currently stands. Therefore, we invite you to submit a revised version of the manuscript that addresses the points raised during the review process.

The authors should carefully attend to the suggestions about clarifications or improvements to the methodology and results. Key focal points are conducting a parallel trends test for the difference in difference (reviewer 2), clarifying the analysis of unauthorized parents (reviewer 2), clarifying the description of the results (reviewer 1) and considering adding more limitations of the study in the discussion (reviewer 1 and 2).

We look forward to receiving your revised manuscript.

Kind regards,

Jim P Stimpson, PhD

Academic Editor

PLOS ONE

Journal Requirements:

2.  Please improve statistical reporting and refer to p-values as "p<.001" instead of "p=.000". Our statistical reporting guidelines are available at https://journals.plos.org/plosone/s/submission-guidelines#loc-statistical-reporting

3. According to our submission guidelines (https://journals.plos.org/plosone/s/submission-guidelines#loc-human-subjects-research), for studies involving humans categorized by race/ethnicity, age, disease/disabilities, religion, sex/gender, sexual orientation, or other socially constructed groupings, authors should explicitly describe their methods of categorizing human populations, and define categories in as much detail as the study protocol allows.

4. Please provide additional details regarding participant consent. In the ethics statement in the Methods and online submission information, please ensure that you have specified (1) whether consent was informed and (2) what type you obtained (for instance, written or verbal, and if verbal, how it was documented and witnessed). If your study included minors, state whether you obtained consent from parents or guardians. If the need for consent was waived by the ethics committee, please include this information.

Reviewers' comments:

Reviewer's Responses to Questions

**Comments to the Author**

1. Is the manuscript technically sound, and do the data support the conclusions?

Reviewer #1: Partly

Reviewer #2: Yes

2. Has the statistical analysis been performed appropriately and rigorously? 

Reviewer #1: Yes

Reviewer #2: Yes

3. Have the authors made all data underlying the findings in their manuscript fully available?

Reviewer #1: No

Reviewer #2: Yes

4. Is the manuscript presented in an intelligible fashion and written in standard English?

Reviewer #1: No

Reviewer #2: Yes

5. Review Comments to the Author

Reviewer #1: - I suggest changing the article title—this study is important however, it is important to provide context (this study was done using data from one state/county, and it is from 2005/2006)

- Use past tense (abstract is written in present tense for example—expand past tense throughout entire manuscript, tables, etc.)

Introduction

- There is both an introduction and a background section—I believe these two should be heavily revised and condensed to one introduction section

- Suggestion for possible paragraphs:

o Immigration (what does this look like in the United States? who is immigrating to the United States? Discuss immigration in North Carolina as this study relies on data from North Carolina—give us the context), discuss different stratifications (e.g., citizenship, documentation status), immigration versus immigrant policies and enforcement (federal versus state level)—this is important to discuss and differentiate; just because 287(g) is in the books, this does not always equate to enforcement so this is important to discuss clearly

o Access to health care for immigrants/mothers/their US born children and the impacts of immigrant policies; what do these disparities look like in North Carolina (a state with high levels of criminalization and low levels of integration immigrant policies)

Exclusionary policies, which entail enforcement, influences eligibility and access to programs and avoidance of human services (e.g., fear)

o What does enforcement do for maternal stress and health care access for themselves AND their children/families? Discuss this relationship clearly

Suggestions for references to include:

• Health Care Access and Utilization for Latino Youth in the United States: The Roles of Maternal Citizenship and Distress

• Inclusive state immigrant policies and health insurance among Latino, Asian/Pacific Islander, Black, and White noncitizens in the United States

• Included, but deportable: A New public health approach to policies that criminalize and integrate immigrants

• A social determinants framework identifying state-level immigrant policies and their influence on health

- In revising and condensing the introduction and background sections into one introductory section, I suggest clearly laying the study framework (mini outline above that may be helpful)—what leads to what, what is associated with what, and so on—there has to be clearer discussion of these pathways/relationships/associations that lead from ICE enforcement to birthweight disparities.

- Line 37-39—there are differences between stress, stresses, stressors –please clarify/revise this sentence

- Line 116-118—how is this related to enforcement? If there is reduced trust in medical providers? How is this related? Explain this or remove this statement. This builds again, on the need for a clear study framework.

Methods

- Prior to implementation of 287(g), what was the context/media coverage for this? Undocumented immigrant populations are heavily influenced by word of mouth/media coverage of immigrant policies—just the notion of 287(g) being a possibility will instill fear and stress among undocumented immigrant populations (related to the robustness check described line 229). I would disagree that these findings would “cast doubt on policy-induced impact on outcomes”—because this population is immensely marginalized and vulnerable and just as we have seen with discussions of the public charge regulation under Trump, families were not accessing programs/benefits for their children who may have been entitled to these programs/benefits, and the disparities only got wider when the public charge regulations were officially implemented. Just discussion of these exclusionary policies influences the health and well-being of immigrant families, especially those with undocumented members.

- Line 242—do you mean Hispanic US-born parents instead of less-educated foreign-born parents?

- For the triple diff-in-diff, did you consider instead estimating the double-difference in the two separate subsamples?

- Line 263-265—what does “federal level context” mean here? Can you give an example of what exactly you are saying here?

Results

- Line 293-295—compared to who?

- Line 325—these results should be included as a supplement given the impact having two parents makes to the economic and social environment for mothers and their babies

- Line 327-329—this explanation is not clear—better educated foreign-born parents are more likely residing in the US with some documentation status (e.g., certain visas, residency, etc.) and this is associated with higher income/higher economic resources compared to someone who is undocumented. I do not believe this explanation is an “or” statement, but more so there is a pathway between documentation/citizenship status and social/economic conditions for immigrant populations—this needs to be stated and understood the authors.

- Some of the information in the results section belongs in the discussion section—results only presents results and not explanations or possible theories for why the results are what they are

- Line 343-345—I caution the confidence and strong language used with respect to falsification tests—it is difficult to evaluate with high certainty that the point estimates are not biased.

- Line 356-358—clarify/specify if you are or are not comparing less educated foreign born to less educated US-born

- Line 374—how did the authors compare their findings? By just researching other similar studies or was some sort of meta-analyses done?

- Line 374-382—I believe these findings may be more comparable to discrimination literature for Black mothers and low birthweight babies—the examples used in these lines are from extreme acute events—policy contexts and social contexts that lead to certain (e.g., exclusionary) policies being enacted are not an overnight event. Undocumented immigrants in contexts where 287(g) would be implemented are already experiencing exclusionary contexts/environments—these are long term effects that are compounded/worsened by these policies—these policies are essentially brewing in these areas and the stressors of these environments/contexts is deep for these populations, leading to hypervigilance/high stress over time (chronic)—not only after 287(g) is implemented—I highly suggest removing these lines as they are misleading and negate long term exposure to racism and discrimination in these environments that leads to these exclusionary policies that are enacted.

- Lines 383-390—again—WIC would not cancel out the harm of 287(g) because undocumented immigrant mothers may not be entitled to WIC in a state—this paragraph should be removed or updated to reflect the context in the county with which this data is from.

- Can the authors provide results from their parallel trends of the outcomes pre-287(g)?

- Did the authors consider possibly applying different methodological approaches to understand the complex pathway to understand the impact of 287(g)? for instance, how much is having a low birth weight baby attributed to 287(g) versus to low utilization of prenatal care? (mediation is possibly occurring – lines 440-441)

Discussion

- Results section should be heavily revised and some of the points are more fitting for the discussion section

- Limitations section is missing several points, including not having citizenship/documentation status information; maternal insurance coverage status, prenatal care/provider concentration ratios (are there adequate providers for where these mothers reside?)

Reviewer #2: This is an extremely well written article on an important and timely topic. The design and analyses are well suited to the research question, and thoroughly carried out. I commend the authors for this submission. The background is thorough, and does an excellent job of describing the policy import at the same time that it reviews the underlying pathophysiologic mechanism (not an easy thing to do succinctly!). The methods are thorough and clear and the results clearly described. I offer a few minor points as considerations for improvement.

- Line 141 refers to ‘’detainers” but does not define this term. One can surmise its meaning by reading further down but an explanation up front would help

- Line 257 and the first sentence of the next paragraph seem to be contradictory - were counties that adopted 287(g) programs after Mecklenburg included? The way it is written it seems like they were included in line 257 but then were not included in the next paragraph.

- Line 329 refers to unauthorized parents – this made me pause as I thought I had missed an analysis of unauthorized parents. Could the authors signal to the reader that this is speculation/assumed, or something to indicate to the reader that these analyses were not done as part of this paper?

- One limitation to consider is that people can live in one county and work in another. So, residents of nearby counties might have been affected (are these included?) which would lead to underestimation of treatment effect. I would imagine that residents of Mecklenburg who work in another county would be affected either way as they would need to move within the county.

- Did the authors test the parallel trends assumption of difference in difference analyses? If not, why not?

- Could the authors include an economic estimate in the discussion? Are there estimates of lifetime costs associated with decreases in birthweights for example? This would strengthen the study greatly.

6. PLOS authors have the option to publish the peer review history of their article (what does this mean?). If published, this will include your full peer review and any attached files.

Reviewer #1: No

Reviewer #2: No

---

## [Author Response · Author response to Decision Letter 0]

11 Dec 2020

December 6, 2020

Dear Professor Stimpson,

Thank you for the invitation to submit a revision of our paper along the lines suggested in the referee reports and your suggestions. We appreciate this opportunity. We also thank you and the two referees for their constructive comments.

Below, we provide our responses to the you and to the two reviewers. Your comments, as well as those of the Reviewers, are in italics, with our responses are in plain text. At the end of the letter, we discuss the sharing of our data set, as required by PLOS ONE.

We hope that in addressing these comments, that the manuscript is now suitable for publication in PLOS one.

Sincerely,

Christina Gibson-Davis, Romina Tome, Marcos A. Rangel and Laura Bellows

Responses to Comments from the Editor

The authors should carefully attend to the suggestions about clarifications or improvements to the methodology and results. Key focal points are:

a) conducting a parallel trends test for the difference in difference (reviewer 2), 

Parallel trends tests are discussed in the text (starting at line 201 and description of results starting at line 323) and presented in Table 4. 

b) clarifying the analysis of unauthorized parents (reviewer 2), 

We clarify throughout the text that undocumented status is proxied and not exact (lines 103 and 400). We emphasize that we are not aware of administrative data sets with birth outcomes which contain such information. 

c) clarifying the description of the results (reviewer 1) and 

We have tried our best in reorganizing the discussion of results and the order in which it occurs on the text. We now present results plain and simple (line 250) and devote a section to their self-contained discussion (line 357). 

d) considering adding more limitations of the study in the discussion (reviewer 1 and 2).

We have included in this revised version an entire section on Limitations (line 397) right before the conclusion and after results are presented and discussed. 

Sharing of Data

We are unable to share the data as the owner of the data, the North Carolina Department of Vital Statistics, prohibits the sharing of birth record data that contains geographic identifiers. As our data contains a geographic identifier (e.g., county of residence), we are unable to share it. Users wishing to obtain the data should contact the department at 919-733-3000 (email addresses are not available to the public).

 

Responses to Reviewer #1

Thank you for your detailed comments and suggestions. Below, we provide our responses to your comments, including text from your report (in italics), and our responses (in plain text). Each of our responses contains a description of the changes introduced in the manuscript to address your comments and located them in the main text (in brackets).

1. I suggest changing the article title—this study is important however, it is important to provide context (this study was done using data from one state/county, and it is from 2005/2006)

Thank you. Following your advice, we have modified the title of the paper to “Heightened immigration enforcement impacts US citizens’ birth outcomes: evidence from early ICE interventions in North Carolina.”

[see: cover]

2. Use past tense (abstract is written in present tense for example—expand past tense throughout entire manuscript, tables, etc.)

Thanks for your observation. We have updated the abstract and entire manuscript to use past tense whenever deemed appropriate. In doing so we use the journal’s guidelines to authors and, in particular, the following general directions: 

a) The present tense is used for general facts

b) The present tense is used when the article is either the subject of the sentence or the thing to which you are referring 

c) When talking about an actual observation the past tense is used. 

[see: whole text]

3. There is both an introduction and a background section—I believe these two should be heavily revised and condensed to one introduction section.

Suggestion for possible paragraphs:

o Immigration (what does this look like in the United States? who is immigrating to the United States? Discuss immigration in North Carolina as this study relies on data from North Carolina—give us the context), discuss different stratifications (e.g., citizenship, documentation status), immigration versus immigrant policies and enforcement (federal versus state level)—this is important to discuss and differentiate; just because 287(g) is in the books, this does not always equate to enforcement so this is important to discuss clearly

o Access to health care for immigrants/mothers/their US born children and the impacts of immigrant policies; what do these disparities look like in North Carolina (a state with high levels of criminalization and low levels of integration immigrant policies)

§ Exclusionary policies, which entail enforcement, influences eligibility and access to programs and avoidance of human services (e.g., fear)

o What does enforcement do for maternal stress and health care access for themselves AND their children/families? Discuss this relationship clearly

§ Suggestions for references to include:

• Health Care Access and Utilization for Latino Youth in the United States: The Roles of Maternal Citizenship and Distress

• Inclusive state immigrant policies and health insurance among Latino, Asian/Pacific Islander, Black, and White noncitizens in the United States

• Included, but deportable: A New public health approach to policies that criminalize and integrate immigrants

• A social determinants framework identifying state-level immigrant policies and their influence on health

In revising and condensing the introduction and background sections into one introductory section, I suggest clearly laying the study framework (mini outline above that may be helpful)—what leads to what, what is associated with what, and so on—there has to be clearer discussion of these pathways/relationships/associations that lead from ICE enforcement to birthweight disparities.

Thanks for these detailed suggestions and guidance. We have addressed this comment by combining the Introduction and Background sections in one longer introduction section. In this updated introduction, we add details about the 287(g) program; immigration in North Carolina; pathways connecting 287(g) program, maternal stress, health at birth, and prenatal care utilization. We have also added your suggested references to this new section. We have done so, however, also trying to keep up with space limitations. 

[see: Introduction]

4. Line 37-39—there are differences between stress, stresses, stressors –please clarify/revise this sentence

Thank you for raising this point. To avoid confusing terms, we edited the manuscript to include only stress and its plural, stresses.

[see: whole text]

5. Line 116-118—how is this related to enforcement? If there is reduced trust in medical providers? How is this related? Explain this or remove this statement. This builds again, on the need for a clear study framework.

Thank you for this comment. In the revised text, this statement has been deleted. This is not our study’s framework but we use it to motivate our data analysis as it suggests a potential channel of operation of impacts. We cannot examine this mechanism directly because we designed an inferential study. It is based on reasoning generated by findings documented in Rhodes et al. (2015) [ref #31]. In their article they documented, using focus groups, that after the implementation of ICE’s programs in North Carolina, Hispanic mothers did not trust staff at agencies providing services. These mothers worried to be detained or deported if they visit health clinics without documentation, and they also reported concerns about exacerbating anti-immigrant sentiments and racial profiling and discrimination, including within health care settings. The work we mention here is still cited on lines 65, 92 and 384, nonetheless. 

[see: NA - deleted statement]

6. Prior to implementation of 287(g), what was the context/media coverage for this? Undocumented immigrant populations are heavily influenced by word of mouth/media coverage of immigrant policies—just the notion of 287(g) being a possibility will instill fear and stress among undocumented immigrant populations (related to the robustness check described line 229). I would disagree that these findings would “cast doubt on policy-induced impact on outcomes”—because this population is immensely marginalized and vulnerable and just as we have seen with discussions of the public charge regulation under Trump, families were not accessing programs/benefits for their children who may have been entitled to these programs/benefits, and the disparities only got wider when the public charge regulations were officially implemented. Just discussion of these exclusionary policies influences the health and well-being of immigrant families, especially those with undocumented members.

We appreciate your comment and suggestion. As you clearly state, media coverage and potential anticipatory effect would be a problem for our estimation strategy. To address this concern, we have highlighted in the text that our falsification exercise included in the paper originally submitted can examine if anticipatory effects are present. In this test, designed to detect parallel- trends between treatment and control locations, we use data from 2004-2005 and assume that the 287(g) program was implemented one year before it actually was. Results from this exercise recover pre-treatment trends that would also be an indicative of an anticipation effect. We show in the first column of Table 4 that this is not the case in our study. In order to clarify this point, we have added a longer explanation of these test in the Analytical Strategy section. 

However, as we do not observe media coverage before the introduction of the 287(g) program in Mecklenburg, we have added this concern as one of our limitations in our new Limitations section. 

[see: section Analytical Strategy, line 207 ; section Limitations, line 422]

7. Line 242—do you mean Hispanic US-born parents instead of less-educated foreign-born parents?

Thank you for raising this point. In the triple difference model that we were describing in the referred lines of the original submission, we compare less-educated foreign-born parents and non-Hispanic US-born parents. That is, we compare the main group of parents in our analysis with the sample of US-born parents. In order to clarify this point in the paper, we have corrected this sentence. Specifically, we have replaced the text “The second compared the difference-in-differences estimates for less-educated foreign-born parents to the difference-in-differences estimate for non-Hispanic US-born parents.” by “The second compared the difference-in-differences estimates for our sample of interest, less-educated foreign-born parents, to the difference-in-differences estimate for non-Hispanic US-born parents.” 

[see: line 222]

8. For the triple diff-in-diff, did you consider instead estimating the double-difference in the two separate subsamples?

Thanks for this comment and we apologize for the unclear text. We have estimated the double-difference in separate subsamples. These results are reported in Table 3 (third to fifth columns). Then, we reported (in Table 4 and 5) the results from the triple diff-in-diff model because it provides not only the coefficient of the difference between estimates from each difference-in-difference but also the standard errors of this coefficient. So, for example, the difference-in-difference estimates for birth weight, we report an effect of -58.02 for births to less-educated foreign-born parents (Table 3, column 3) and an effect of 0.52 for births to non-Hispanic US-born parents (column 5). The difference between these two estimates is -58.54; this estimate is what we report in Table 5 (first column). We hope that this discussion is clearer in the revised text

[see: Tables 3, 4 and 5 and companion discussion]

9. Line 263-265—what does “federal level context” mean here? Can you give an example of what exactly you are saying here?

Thank you for raising this point. What we mean in that sentence is that policies which are adopted locally are likely to be also responding to the overall policy environment being pushed by the federal administration’s Department of Homeland Security. Then, if ICE learns from early implementations (anywhere in the country), that affects and, probably changes the way, ICE operates in future implementations of the program. Following your comment, we have edited the last paragraph of the Analytical Strategy section to include that clarification. 

[see: line 240]

10. Line 293-295—compared to who?

Thank you for your comment. In this sentence we compare less-educated foreign-born mothers who reside in Mecklenburg County with similar mothers who live in other counties. To make this sentence clearer we have replaced the text “Utilization of prenatal care, however, was lower in Mecklenburg County relative to elsewhere for less-educated foreign-born mothers.” to “Utilization of prenatal care for less-educated foreign-born, however, was lower in Mecklenburg County relative to similar parents elsewhere even before the policy change we evaluate. Since our empirical strategy is based on time-changes, this difference in levels is unlikely to invalidate our results and conclusions, as further discussed below.”

[see: line 275]

11. Line 325—these results should be included as a supplement given the impact having two parents makes to the economic and social environment for mothers and their babies

Following your suggestion, we have added the results from estimating the difference-in-differences model for the different samples of births taking into account mothers’ demographics (rather than both parents’ characteristics) in S2 Table. We show in this table that the estimated effect is lower for the sample of births to less-educated foreign-born mothers than for the sample of births to less-educated foreign-born parents, as expected considering that less-educated immigrants, and men in particular, are the groups most likely to experience removal or fear of removal (Pham, 2018). 

[see: Table S2]

12. Line 327-329—this explanation is not clear—better educated foreign-born parents are more likely residing in the US with some documentation status (e.g., certain visas, residency, etc.) and this is associated with higher income/higher economic resources compared to someone who is undocumented. I do not believe this explanation is an “or” statement, but more so there is a pathway between documentation/citizenship status and social/economic conditions for immigrant populations—this needs to be stated and understood the authors.

Thanks for your comment. We agree that the lack of policy impact on better educated parents could reflect both their likely access to additional economic resources and that they are less likely to be unauthorized, and we believe this text was a poorly written rather than a lack of understanding on that point (as presumed by the reviewer). Indeed, we had addressed these two conditions in the Discussion section in our initial submission. However, as part of our response also to your next comment, we have deleted this statement from the Results sections, and we only left the related statement in the Discussion section of the revised manuscript. Specifically, we explain the following in the third paragraph of the Discussion section: “We reason that effects are concentrated among less-educated parents for two main reasons. First, given assortative mating, women without high school diplomas are likely to have partners with similar levels of education [53]. Insofar as less-educated individuals are more likely than better-educated individuals to be unauthorized [41] and therefore directly subject to 287(g) programs’ deportation risk. Less-educated mothers may be more aware of this targeting and experience greater levels of stress when fathers are potential targets. Second, even if they are equally impacted by stress and anti-immigrant sentiment in the location where they reside, better-educated women are more likely to have access to health-protective economic resources.”

[see: line 387]

13. Some of the information in the results section belongs in the discussion section—results only presents results and not explanations or possible theories for why the results are what they are

We have edited the Results section to address this suggestion. In the revised manuscript, we describe results in this section, and we have moved any further explanation or discussion about them to the Discussion section. Specifically, we moved, from the Results section to the Discussion section, the explanation about the lack of effects on better-education foreign-born parents and a paragraph comparing our estimates to results in other studies. We also removed from the Results section the last paragraph in the initial submission that described the limitations of our study, and we included an edited version of this paragraph in our new Limitations section. Lastly, in this revision we also included a Conclusion section. 

[see: Results; Limitations; Conclusion]

14. Line 343-345—I caution the confidence and strong language used with respect to falsification tests—it is difficult to evaluate with high certainty that the point estimates are not biased.

Thank you for your comment. We now refer to our robustness exercises as not finding evidence to invalidate the parallel trends assumption. Parallel trends tests are discussed in the text and presented in Table 4. 

[see: lines 201, and 324; estimates in Table 4]

15. Line 356-358—clarify/specify if you are or are not comparing less educated foreign born to less educated US-born

In our triple-difference model, as well as in the other specification, we consider the sample of (all) non-Hispanic US-born parents. That is defined regardless of their level of education. Then, the triple-difference model, described in the lines 356-358 of our original submission, compares estimates for less-educated foreign-parents and for (all) non-Hispanic US-born parents. In order to make these sentences clearer, we have edited the text in the Results section that describes Table 5. 

[see: line 338]

16. Line 374—how did the authors compare their findings? By just researching other similar studies or was some sort of meta-analyses done?

Thanks for your question. In our initial submission, to compare our findings with other studies in the literature, we chose similar studies exploring the effect of shocks on health at birth. In the revised manuscript, following your next comment, we have removed the paragraph comparing our results to articles studying the consequences of shocks in utero. Moreover, we have edited and moved the comparison with changes in maternal nutrition to the Discussion section following your suggestions. And to be clear, no, a meta-analyzes was not conducted by the authors since it was not the objective of our program of study. 

17. Line 374-382—I believe these findings may be more comparable to discrimination literature for Black mothers and low birthweight babies—the examples used in these lines are from extreme acute events—policy contexts and social contexts that lead to certain (e.g., exclusionary) policies being enacted are not an overnight event. Undocumented immigrants in contexts where 287(g) would be implemented are already experiencing exclusionary contexts/environments—these are long term effects that are compounded/worsened by these policies—these policies are essentially brewing in these areas and the stressors of these environments/contexts is deep for these populations, leading to hypervigilance/high stress over time (chronic)—not only after 287(g) is implemented—I highly suggest removing these lines as they are misleading and negate long term exposure to racism and discrimination in these environments that leads to these exclusionary policies that are enacted.

Thank you for your suggestion. To address it, we have removed the paragraph comparing our results to studies examining the effect of different shocks during the time in utero. We did not add any literature about the discrimination literature for Black mothers, as we respectfully disagree that this literature is relevant for a policy change such as the introduction of 287(g). Though policy changes such as 287(g) may have long antecedents and consequents (as you suggest), we do not think that they are comparable to the multi-generational, cumulative detrimental effects of racism. If the Editor disagrees with this decision, we are happy to include these references in a subsequent revision. 

18. Lines 383-390—again—WIC would not cancel out the harm of 287(g) because undocumented immigrant mothers may not be entitled to WIC in a state—this paragraph should be removed or updated to reflect the context in the county with which this data is from.

After reading this comment and reviewing the manuscript we realized that our idea regarding WIC was not clear. We reported the effects of changes in maternal nutrition through the food stamp program and WIC to put our results in context. We were not implying that using WIC would cancel out the impact of the 287(g) program. But, as an aside and just to make sure the reviewer understands, WIC eligibility is not limited to documented immigrants – undocumented immigrants are not explicitly excluded from the program in any of the 50 US states. In order to clarify the point we were trying to make on the text, we have edited the paragraph making this comparison that is in the Discussion section in the revised manuscript (following your comment about editing the content of the Results section).

[see line 363] 

19. Can the authors provide results from their parallel trends of the outcomes pre-287(g)?

We now refer to our robustness exercises as not finding evidence to invalidate the parallel trends assumption. Parallel trends tests are discussed in the text (starting at line 201 and description of results starting at line 323) and presented in Table 4. 

[see: lines 201, and 324; estimates in Table 4]

20. Did the authors consider possibly applying different methodological approaches to understand the complex pathway to understand the impact of 287(g)? for instance, how much is having a low birth weight baby attributed to 287(g) versus to low utilization of prenatal care? (mediation is possibly occurring – lines 440-441)

We considered conducting a mediation analysis to study the pathways of the impact of the 287(g) program. However, preferred to use a reduced-form approach in our investigation. The rational for that choice is that if ICE activities makes low-education foreign-born mother reduce their prenatal care utilization and getting less or later prenatal care affects health at birth, it is still the case that the ICE’s program leads to worse birth outcomes. Then, our estimates invite readers to consider the overall impact of 287(g) program (regardless of channel) and present a non-exhaustive list of potential channels, including pre-natal care. In addition, we cannot rule out that reduced or later pre-natal care may be a marker of other (correlated) channels, and it is not reasonable to attribute all these mediating effects to worse pre-natal care utilization. 

21. Results section should be heavily revised and some of the points are more fitting for the discussion section

Thank you for your suggestion. To address that, we have edited the Results and Discussion sections. Specifically, we moved from the Results section to the Discussion section the explanation about the lack of effects on better-education foreign-born parents (lines 327-329 and lines 361-364 in the initial submission) and a paragraph comparing our estimates to results in other studies (lines 383-390 in the initial submission). We also removed from the Results section the last paragraph in the initial submission that described the limitations of our study, and we included an edited version of this paragraph in our new Limitations section. Lastly, in this revision we also included a Conclusion section. 

[see: Limitations; Conclusion]

22. Limitations section is missing several points, including not having citizenship/documentation status information; maternal insurance coverage status, prenatal care/provider concentration ratios (are there adequate providers for where these mothers reside?)

Thank you for your comment. Regarding your first point about citizenship/documentation status information, we acknowledged the lack of this information in the Data section in the initial submission, and we added it in our Limitations section in the revised manuscript. It is worth mentioning that we are not aware of any population wide administrative data that report documentation status. Also, using both mothers and fathers’ information to define our samples allow us to get a better proxy of undocumented families than other studies using only maternal information because less-educated immigrants, and men in particular, are the groups most likely to experience removal or fear of removal (Pham 2018). 

Regarding your second point, we understand that it would be great to provide that information for the context of the study, but we do not think that maternal insurance coverage status or prenatal care/provider concentration ratios could be a concern for our empirical strategy or limitation. These two conditions would be a concern only if they change at the same time that Mecklenburg implemented the 287(g) program, yet there is no reasons to necessarily expect that. 

 

Responses to Reviewer #2

Thank you for your constructive comments and suggestions, which were very helpful in

the revision. Below, we provide our responses to your comments, including text from

your report (in italics), and our responses (in plain text). Each of our responses contains

a description of the changes introduced in the manuscript to address your comments and suggestions.

1. Line 141 refers to ‘’detainers” but does not define this term. One can surmise its meaning by reading further down but an explanation up front would help

Thank you for your comment. We have edited the text to clarify this point. Specifically, we added the following definition for detainers: “i.e., documents that allow local law enforcement agencies to hold immigrants until they can be placed in removal proceedings.”

[See: line 114]

2. Line 257 and the first sentence of the next paragraph seem to be contradictory - were counties that adopted 287(g) programs after Mecklenburg included? The way it is written it seems like they were included in line 257 but then were not included in the next paragraph.

Thank you for raising this point. We have clarified both paragraphs. The first statement refers to using alternative definitions to select counties in the control group as a robustness check (line 257 in the initial submission) while we keep the same treatment group (Mecklenburg County). The second paragraph refers to the justification of not including counties that adopt 287(g) programs later as part of our treatment group. 

[See: lines 236 and 240]

3. Line 329 refers to unauthorized parents – this made me pause as I thought I had missed an analysis of unauthorized parents. Could the authors signal to the reader that this is speculation/assumed, or something to indicate to the reader that these analyses were not done as part of this paper?

Thank you for this suggestion. As you stated, along the manuscript we refer to “likely unauthorized parents” because our data do not report documentation status (as any administrative datasets). Then, better-educated parents are less likely to be unauthorized. Following a suggestion made by Reviewer #1, we have cut the sentence referring to this point in the Results section, yet we carefully explain the different effects found for less and better educated foreign-born parents in the Discussion section addressing your point. 

[See: line 375]

4. One limitation to consider is that people can live in one county and work in another. So, residents of nearby counties might have been affected (are these included?) which would lead to underestimation of treatment effect. I would imagine that residents of Mecklenburg who work in another county would be affected either way as they would need to move within the county.

Thank you for your comment. We lack information about whether and where parents work, so we cannot provide evidence about this concern, yet counties around Mecklenburg are included in our control group. So, we have added this point as one of our limitations. Because the list of limitations in the revised manuscript includes more points, we have moved it to a new Limitations section.

[See; Limitations section]

5. Did the authors test the parallel trends assumption of difference in difference analyses? If not, why not?

Thank you for this point. We have conducted and report the findings. We refer to our robustness exercises as not finding evidence to invalidate the parallel trends assumption. Parallel trends tests are discussed in the text (starting at line 201 and description of results starting at line 323) and presented in Table 4. 

[see: lines 201, and 323; estimates in Table 4]

6. Could the authors include an economic estimate in the discussion? Are there estimates of lifetime costs associated with decreases in birthweights for example? This would strengthen the study greatly.

We have included a paragraph on the Discussion session making reference to previous literature on these effects. 

[see: line 375]

 

References

Pham H. 287(g) Agreements in the Trump Era. Wash Lee Law Rev. 2018;75(3):1253–1286. https://scholarlycommons.law.wlu.edu/wlulr/vol75/iss3/3/

Rhodes SD, Mann L, Simán FM et al. The impact of local immigration enforcement policies on the health of immigrant Hispanics/Latinos in the United States. Am J Public Health. 2015;105(2):329–337. doi:10105/AJPH.2014.302218

---

## [Decision Letter · Decision Letter 1]

21 Dec 2020

Heightened immigration enforcement impacts US citizens’ birth outcomes: Evidence from early ICE interventions in North Carolina

PONE-D-20-30120R1

Dear Dr. Gibson-Davis,

We’re pleased to inform you that your manuscript has been judged scientifically suitable for publication and will be formally accepted for publication once it meets all outstanding technical requirements.

Kind regards,

Jim P Stimpson, PhD

Academic Editor

PLOS ONE

Reviewer's Responses to Questions

**Comments to the Author**

1. If the authors have adequately addressed your comments raised in a previous round of review and you feel that this manuscript is now acceptable for publication, you may indicate that here to bypass the “Comments to the Author” section, enter your conflict of interest statement in the “Confidential to Editor” section, and submit your "Accept" recommendation.

Reviewer #1: All comments have been addressed

2. Is the manuscript technically sound, and do the data support the conclusions?

Reviewer #1: Yes

3. Has the statistical analysis been performed appropriately and rigorously? 

Reviewer #1: Yes

4. Have the authors made all data underlying the findings in their manuscript fully available?

Reviewer #1: Yes

5. Is the manuscript presented in an intelligible fashion and written in standard English?

Reviewer #1: Yes

6. Review Comments to the Author

Reviewer #1: (No Response)

7. PLOS authors have the option to publish the peer review history of their article (what does this mean?). If published, this will include your full peer review and any attached files.

Reviewer #1: No

---

## [Editor Report · Acceptance letter]

8 Jan 2021

PONE-D-20-30120R1 

Heightened immigration enforcement impacts US citizens’ birth outcomes: Evidence from early ICE interventions in North Carolina 

Dear Dr. Gibson-Davis:

I'm pleased to inform you that your manuscript has been deemed suitable for publication in PLOS ONE. Congratulations! Your manuscript is now with our production department. 

Kind regards, 

on behalf of

Dr. Jim P Stimpson 

Academic Editor

PLOS ONE